# Design of a Multimodal Oculometric Sensor Contact Lens

**DOI:** 10.3390/s22186731

**Published:** 2022-09-06

**Authors:** Jean-Louis de Bougrenet de la Tocnaye, Vincent Nourrit, Cyril Lahuec

**Affiliations:** 1Optics Department, Institut Mines-Télécom Atlantique, Technopôle Brest Iroise, CS 83818, CEDEX 03, 29238 Brest, Brittany, France; 2Laboratoire des Sciences et Techniques de l’Information, de la Communication et de la Connaissance, CNRS UMR 6285, 29238 Brest, Brittany, France

**Keywords:** smart contact lens, eye-tracking, pupil size, accommodation, diffractive optical element

## Abstract

Oculometric data, such as gaze direction, pupil size and accommodative change, play a key role nowadays in the analysis of cognitive load and attentional activities, in particular with the development of Integrated Visual Augmentation Systems in many application domains, such as health, defense and industry. Such measurements are most frequently obtained by different devices, most of them requiring steady eye and body positions and controlled lighting conditions. Recent advances in smart contact lens (SCL) technology have demonstrated the ability to achieve highly reliable and accurate measurements, preserving user mobility, for instance in measuring gaze direction. In this paper, we discuss how these three key functions can be implemented and combined in the same SCL, considering the limited volume and energy consumption constraints. Some technical options are discussed and compared in terms of their ability to be implemented, taking advantage of recent developments in the field.

## 1. Introduction

Different multimodal features related to peripheral physiology, brain activity, and oculometric reactions have been used as non-intrusive, reliable, and objective measures of cognitive load and attention [1]. In this paper, we focus on three of them: gaze direction (GD), pupil dilation (PD) and accommodative power (AP). A blinking measurement can also be easily added [2]. Numerous works have shown that these oculometric features provide complementary information about the subject’s cognitive load and attention, and yield great benefits when combined. Furthermore, the solutions used to implement them share some common elements that could be mutualized. This emphasizes their ability to be brought together, and then be miniaturized to become portable and provide the user with full mobility. In addition, the measurement of most of them demands the control of eye motion and the lighting conditions [3]. With the development of Integrated Visual Augmentation Systems (IVASs), these approaches could easily be incorporated into said systems to provide accurate and reliable data, regardless of eye movement and the wearer’s motions. In consequence, the eye is the best location for their integration.

The use of smart contact lenses (SCL) in human–machine interactions (HMIs) has received attention recently with the joint development of IVAS and nanotechnologies [4]. For instance, the Sony blink-powered SCL records everything the user sees, using a picture-taking unit, and a piezoelectric sensor capable of detecting the difference between voluntary and involuntary blinks [5]. California University developed a robotic telescopic SCL to zoom in on objects of interest by blinking [6]. Among the visual HMIs [7], eye-trackers have become some of the most popular interfaces for assessing and modulating sensorimotor and cognitive functions. For instance, they have been used for a wide variety of tasks, from the most basic, such as selection, manipulation and navigation [8,9], to assessing mental workload [10], passing by probing memory [11] and analyzing programming technologies [12]. 

In the future of IVAS, eyes will replace standard tools such as cursors, touch-screens, touchpads and keyboards to convey visual intents or commands, and to identify cognitive loads. Therefore, a reliable control of oculometric parameters is a prerequisite when using the eye as an HMI to control where the user’s attention is focused, or to perform visual designations, thus reducing the operators’ workload and allowing them to concentrate on their main mission, while establishing a new link between planning, control functionalities, and sensory coordination [13]. More generally, such devices are a key element for investigating the impact of multitasking, which is characteristic of the use of mixed reality systems, on our cognitive performance, in combination with other cost-effective Brain–Computer Interfaces (BCIs), such as those described in [14].

We propose here to analyze the various ways to implement gaze direction, accommodative power and pupil size measurements, all of which are operations that are worth implementing on the same contact lens. We recently developed an eye-tracker SCL, and the purpose of this study is to develop a way to measure pupil size and accommodative power simultaneously, while optimizing the number of embedded circuits, the technical complexity, the manufacturing ability and the energy consumption.

## 2. Implementation in an SCL

In this section, we discuss how these functions can be implemented in an SCL. Two main options can be considered. Each sensor delivers specific data (i.e., angle of sight, pupil size, accommodation). These data are either computed at the SCL level and then transmitted via a data link to an external device monitoring the SCL (e.g., eyewear, AR headsets etc.), or used to activate sensors mounted in this monitoring device, which extracts and processes the information at this level. Both options have advantages and drawbacks, which are discussed here. Another obvious requisite for SCL is steadiness when on the eyes. This is obtained using a scleral contact lens, which has the particular features of being stable on the eye and having no direct contact with the cornea. Finally, the SCL has its own energy resource, enabling the on-board functions to operate and wirelessly communicate [15], activate, deactivate, and transfer data to the monitoring device (e.g., goggles, mixed reality headset, etc.).

### 2.1. Eye Gaze SCL

We have developed two prototypes that implement this function [16,17]. Both make use of an instrumented SCL and eyewear. In the first case, shown in Figure 1a, the lens integrates four photodiodes (PTD) illuminated by an infrared source placed on the glasses [16]. The amount of light reaching each photodiode varies with the direction of gaze, and thus this direction can be calculated from the centroid of the PTDs currents. The lens is embedded with an ASIC to perform this calculation, as well as a transmission module to send this data to a sensor on the eyewear. A demonstration with a mock-up achieved 0.11° accuracy with 0.01° precision, which is almost an order of magnitude better than the results achieved by the best commercial head-mounted eye trackers. The secondary antenna was not in the SCL, as seen Figure 1a. In the second case, shown in Figure 1b, the light source was placed within the SCL [17], and the sensor was on the eyewear. More precisely, the SCL contains a VCSEL, which allows the measurement of the direction of gaze with 0.03° accuracy. The detection of the beam can be achieved using a camera or a PSD (which has advantages in terms of integration and speed). In this case, the SCL incorporated a secondary antenna.

### 2.2. Pupillometer SCL

This model is a natural extension of the previous. An SCL including at least one light source, preferably a laser, is combined with an optical detector, allowing the measurement of pupillary diameter. This SCL is coupled with a device for measuring the pupil diameter. The on-board source provides light reflected by the iris surface and analyzed either at the level of the SCL or externally by a suitable device. The pupil size measurement can be done without the voluntary participation of the subject. The novelty lies in the nature of the iris illumination approach [18], the method of reflected beam detection, and especially in the fact that these operations are embedded in an SCL that communicates wirelessly. In contrast to the previous embodiments [16,17], the light source and the optical sensor are oriented towards the retina, pointing at the iris. They are assembled on the inner surface of the electronic circuit. The resourcing and communicating antenna can be arranged on the same side of the periphery of the circuit, as shown in Figure 1b. As said above, such a device allows two distinct modes of implementation.

Both the emission source and the detector (i.e., a photodiode) are in the SCL. In this case, a variation in light intensity is measured (reflected by the portion of the pupil illuminated by the light source—the rest is absorbed in the interior of the eye). A look-up table makes it possible to determine the pupil diameter. This value is computed within the contact lens and transmitted to the eyewear via a conventional NFC link, as in [15]. The principle is illustrated in Figure 2.

The device uses dedicated NXP or ST chips to construct a sensor powered by an NFC protocol. They contain a microcontroller and an ADC converter, and allow energy recovery and data transmission via NFC [15]. In Figure 2, the sensor is on the opposite side of the pupil with respect to the light source, but they could also be placed side by side. The pupil diameter varies between 2 and 8 mm. The light source should illuminate a section of the iris between Rmin and Rmax with, for example, Rmin = 1.5 mm and Rmax = 3 mm. In order to allow the light source to illuminate this entire area, it may be worthwhile to lengthen the optical path. For example, the light source is directed outwards and guided by multiple reflections, as described in [19]. Beam shaping can also be considered using micro-optics, as described in [20], to better illuminate the iris’ arc of interest. The laser beam’s direction can be accurately controlled as well, as shown in [20].

**Figure 2 sensors-22-06731-f002:**
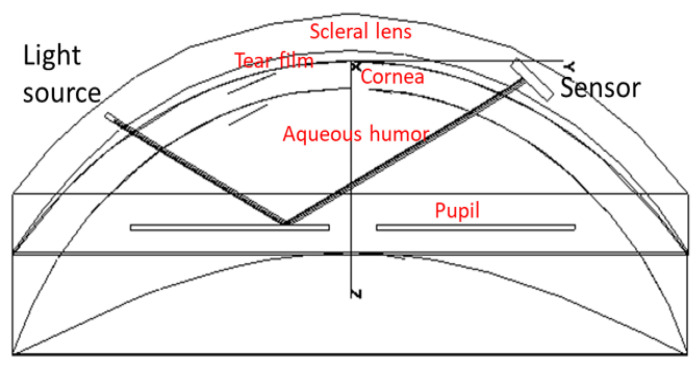
Datacom-based solution: ray-tracing taking into account the different optical interfaces.

The second solution combines a light source and a diffractive optical element (DOE) [20,21], which covers a given area of the iris. This optical element will therefore be partially illuminated with respect to the proportion of light that is reflected from the portion of the iris underlying it (as detailed later on). For this purpose, the DOE [22,23] consists of several facets, each of which creates a light spot at a given distance from the lens, where detectors are placed to determine the diameter of the iris opening. This implementation is illustrated in Figure 3a (operation) and Figure 3b (obtained ray-tracing).

A possible mode of implementation involving an external detection system is based on the use of a micro-optics array (blazed gratings or off-axis Fresnel lenses array, as shown in Figure 4) as the DOE. The reflection on the semi-diffusing surface of the iris has the effect of reducing the coherence of the laser source (a VCSEL here); hence, the use of this type of component is appropriate here.

**Figure 3 sensors-22-06731-f003:**
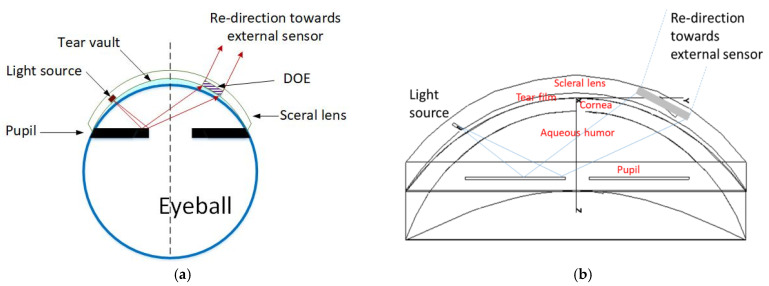
(**a**) Schematic representation of the operation principle: part of the emitted light is reflected by the pupil and detected by a sensor, or impacts a DOE; (**b**) ray-tracing taking into account the different optical interfaces.

For simplicity, the cornea is not represented in the picture; the index of the aqueous humor (1.336), the cornea (1.376) and the lacrimal fluid (1.33) being close, the influence of the cornea refractive power can be neglected if the incidence angle on the cornea is not too large. Hence, the beam reflected from the iris illuminates partially or fully the DOE, depending on the degree of pupil opening. The object focal plane of these lenses is preferably the laser source (not the case in Figure 5). This configuration makes it possible to introduce a 1D detector array (e.g., photodiodes) at a given distance (a few cm), as shown in Figure 6. This figure describes a PRAR SCL (Figure 6a [19]) coupled with an AR headset (Figure 6b), and shows how the liseran IR pointer is detected at the display frame to determine the eye direction and/or activate some headset functions. Here, in our case, the spot detection on each of these detectors allows the pupil size to be derived. The configuration also makes it possible to consider horizontal eye movement (the most relevant) by replacing the 1D array with a matrix of detectors covering a given angular sector (e.g., 15°). 

This solution liberates the choice of the detector array, allowing a better light sensitivity and detector surface. At the SCL level, it simplifies the implementation. There is no need for a photodiode or NXP devices, which is beneficial in terms of energy consumption and volume, keeping the SCL free for embedding other functions. The integration of DOE, which is a flat and passive device, has been tested elsewhere [20].

### 2.3. Autorefractometer SCL

As described in a previous paragraph, various techniques are available to measure the refractive error, most of them relying on recording how light from an infrared source reflects off the retina, and requiring a precise alignment between the infrared source, the eye and the detector. Some of these techniques may be adapted for implementation in an SCL. The first possible means of implementation is described in Figure 7. In this configuration, measurements are achieved at the SCL level. It uses elements already present in the SCL for implementing eye-tracking and pupilometer measurements. It includes a laser that emits light towards the retina, combined with a photodiode embedded in the cavity. The measurement is based on laser feedback interferometry [24]. Basically, some of the reflected light is injected into the cavity. The laser field inside the cavity and the back-injected laser field will be in phase or out of phase, thus modulating the output power via self-mixing interference. If a small fraction of the modulated output power is measured by the photodiode, the phase difference between the beams and thus the eye length can be calculated. This value is then digitized and transmitted via a data link.

In the second of the possible configurations, the laser is combined with a diffractive optical element (DOE) to create one or two spots on the retina. The light reflected from these two spots will create an interference pattern that can be detected by quadratic sensors located around the eye (Figure 8). These detectors are placed around the eye on the eyewear to keep the direction of sight free. In this case, knowledge of the gaze direction will help in compensating for the variable alignment of the eye with the detector. 

In this configuration, measurements are achieved at the eyewear level, similar to the gaze direction solution. Hence, some elements or building blocks could be shared between the AF, the PD and the GD. 

Since we demonstrated [17] that we could send a collimated beam towards the retina, a third option based on beam deviation might be theoretically possible, but the fact that we want to leave the pupil free imposes too many constraints in terms of possible beam paths to make it useful. 

In both cases, laser feedback interferometry (Figure 7) or “classic” interferometry (Figure 8), we propose to use retinal reflection rather than reflections on the crystalline lens surfaces for several reasons. First, the amount of light reflected by the retina will be higher, and the use of interferometry from retinal reflections is a proven technique. Secondly, knowledge of the refractive error can be useful for other applications. Wearing glasses can interfere with the use of many instruments (sighting scope, microscope, augmented reality glasses, virtual reality headset, etc.). If one of these instruments were to incorporate an AF, its optics could be automatically adjusted to the view of each subject, thus allowing them to have a perfect image without wearing corrective glasses.

## 3. Combination of All Sensors in the Same SCL

Except for the differences that arise when the computing task is implemented (inside or outside the SCL), the three functions contain common elements, such as the VCSEL and the antenna harvesting system. Indeed, our SCL eye-tracker can be easily merged with the pupilometer, by implementing a second VCSEL and its optics pointing outside the SCL, to determine the direction of gaze [15,16,17]. Similarly, the suggested SCL refractometer configuration requires the direction of sight be known, and is thus benefited by merging with the laser pointer SCL. Knowledge of the gaze direction of both eyes enables the convergence value to be derived. This information can be efficiently correlated with the accommodation measurements [21].

Furthermore, gaze direction knowledge of each eye permits convergence value determination. The accommodation reflex is the visual response enacted when focusing on nearby objects [21]. Considering that such mechanisms interact normally in a synkinetic way, which consists of the convergence of both eyes, the contraction of the ciliary muscle (accommodation) and pupillary constriction, the measured data are better analyzed correlatively in order to eliminate possible outliers. Alternatively, this approach could also be useful to better determining where the user’s attention is focused when they are exposed to a convergence accommodation conflict (e.g., with 3D). Besides this, each functional design requires at least one laser; these lasers do not need to be activated at the same time, but must be time-multiplexed to reduce the energy consumption, as detailed later on.

With respect to the encapsulation constraint issue, two main parameters should be considered in order to correctly engineer these functions in the same SCL. For all of them, the volume constraint is the most important and prohibitive. The energy consumption should also be considered for many reasons. Even if batteries can be incorporated into the SCL [22], the simplest way to harvest the SCL remains energy transfer by magnetic coupling, which should respect some of the limitations related to ocular safety and SCL heating. 

Furthermore, the criteria are not independent, and can interfere (e.g., embedding a battery will reduce the volume available for other circuits). This point will be discussed in the next section with respect to the alternative implementation, regarding the two parallel computing options.

### 3.1. Proposed Multimodal Oculometric Sensors

In this section, we discuss how an SCL including the three oculometric sensors can be designed according to the two implementation strategies: including or not including data computation at the SLC level. A hydride solution, mixing both strategies, remains possible.

#### 3.1.1. Autonomous Sensor (On-Lens Computing)

On-lens computing can be implemented using a low-power integrated circuit (IC) comprising the necessary blocks required to monitor the sensors (e.g., photodiode and/or interferometer), activate the VCSEL, compute the metrics, transmit the metrics to the eyewear via an NFC link (see [15]), and finally harvest energy,. Off-the-self ICs comprising all the necessary blocks exist. For instance, the NHS3152 from NXP is an NFC-powered IC dedicated to use in smart autonomous sensors. The IC includes a low-power Cortex-M0+ microcontroller, various data converters including a current-to-digital converter (I2D), and a near-field communication and energy-harvester block (power management unit, PMU). Analog and digital ports are also available to connect sensors and actuators. Specific digital pads on the NHS3152 can source up to 20 mA, more than enough to supply the VCSEL. The IC also has I²C and SPI ports to connect smart sensors. The microcontroller can implement a finite-state machine to sequence the lens operation (control the on/off VCSELs (Lasermate VCC-85C10G), measure the photocurrent of the infrared photodiode (Vishay T1170P) to determine the pupil diameter, and communicate with the eyewear or the headset). The NHS3152 consumes about 150 µA under a 1.8 V supply when clocked at 250 kHz, which is fast enough for our application. The 850 nm VCSELs have a threshold current of 0.5 mA. At 850 nm, the relative sensitivity of the T1170P photodiode is 0.95. Figure 9 shows a simplified block diagram of one possible implementation of the multimodal SCL; the supply voltage biasing the photodiode and the VCSELs is provided by the energy harvester. To limit power consumption, the VCSELs are sequentially turned on or off, with a low-frequency (~1 Hz) low duty cycle (<10%) control signal. The VCNL36825T proximity sensor used as a refractometer provides the digitized value of the measured photocurrent, and consumes 100 µA. The dimensions of all the elements are compatible with encapsulation into an SCL as shown in Figure 10. Table 1 gives the dimensions of each part; as seen, the thickest parts, the NHS3152 and the proximity sensor, are only 0.5 mm thick. 

A recent device, proposed by TRUMPF Photonic Components, integrates along a VCSEL (emitting at 850 nm) a photodiode to exploit the principle of self-mixing interference (SMI), allowing the accurate measurement of various parameters, including distances [23]. The 165 × 165 µm^2^ three-pin chip can be connected to the I2D converter using another analogue port of the NHS3152 to measure the photodiode current, thus simplifying the overall design. The small size and low consumption of this device make it compatible with our SCL designs.

A possible implementation on a flexible substrate of the circuit of Figure 10 is shown in Figure 9.

#### 3.1.2. Remote Computing (Off-Lens Computing) 

In this version of the three-function instrumented SCL, none of the metrics are computed in the SCL to simplify the design and fabrication of the encapsulated electronics. Three VCSELs are required: one directed towards the headset as a pointer, and the two others pointing towards the eye, one to form the refractometer, and the other to measure the pupil diameter. As seen in Figure 11, the required electricity need only power the VCSELs and sequence the ON/OFF state of the VCSELs in order to perform the measurements, as was done in [17]. One possible implementation is on a flexible substrate of the circuit shown Figure 12.

### 3.2. Importance of the Addition of DOE

For the second option, the use of optics relays is, in most of the cases, required. As already shown in [20] and suggested in [19], DOEs are used either to collimate or to focus the VCSEL beam when using, for instance, an external detector, which is placed at a given distance from the eye, or to project a pattern at a given distance in front of the eye or on the retina. The DOEs are multi-phase level elements [24], etched into a layer (thickness ~1.8 µm) of spin-coated S1813 photoresist (MicroChem) on 175 µm-thick borosilicate glass substrates using a large custom-built parallel-writing photoplotter [25,26]. Typical DOE diffraction efficiencies of 70–75% are generally observed. The DOE size is determined by the laser beam divergence and the VCSEL to DOE distance, the encapsulation constraint of which is limited to less than 1 mm. Considering our VCSEL divergence (8°) and a distance to the DOE of 680 µm (to limit the lens thickness), the usable surface will be around 200 × 200 µm^2^, as the spot’s diameter is 200 µm. Here, the size of the DOE is 225 × 225 µm^2^ (Figure 13). Only one is observable on this Figure, but a DOE array with four facets, as shown Figure 4, will be easily implemented using this technique. Figure 13 shows the different steps of the integration of the DOE in the SCL, derived from [20]. 

Compared to the on-lens computing case, this option assumes better knowledge of the eye movements in order to provide accurate and reliable oculometric data. Even if the eye motions are in practice limited within a solid angle of less than 15°, the data processing of the various measurements obtained from the different sensor sources by using, for instance, deep learning techniques (including, for instance, knowledge of eye motion common trajectories and saccades) would be useful to make our system more robust.

## 4. Discussion

Based on our encapsulation experience and anticipating future nanotechnology advances, we have proposed and discussed two modes of implementing the three functions in a unique SCL. Such an alternative illustrates the two main methods of using an SCL interfaced with an IVAS. The first one has the advantage of guaranteeing complete mobility, and the capacity to be driven by various devices, such as a smartphone. In contrast, the second option requires a more dedicated monitoring device, such as AR headsets, with the advantage of more sophisticated data detection involving sensor arrays and extended data processing. This option represents the most promising way to use an SCL in combination with an IVAS, such as an AR headset, by using it as a visual cursor to designate commands in menus or targets in 3D scenes. Furthermore, it emphasizes the importance of using a DOE in SCL for many purposes. One of the most challenging, as described in [20], is to be able to directly project on the retina patterns or icons, independently from eye movements and blinking. This is the first step towards what we have called Perivoveal Retinal Augmented Reality (PRAR) [19], which anticipates the transfer of some AR headset display capabilities directly into the SCL, as envisioned for instance by Mojo-Vision [27]. Concerning our main objective, related to the combination in a single SCL of three oculomotor sensors by mutualizing some common elements, we have demonstrated that it is possible with respect to the current state of the art of the technology, and the specific volume and energy constraints of the SCL. Beyond this technological challenge, there is a real benefit to be derived from implementing these functions on the same platform (in particular, the eye itself) considering the near-synkinetic reflex between accommodation, convergence and mydriasis [21]. Furthermore, our multimodal SCL’s ability to record and accommodate eye orientation is nowadays crucial to analyzing mixed reality scenes, where accommodative demand and convergence are often uncoupled. The presence of a gaze pointer in our SCL, detected with an IVAS, will solve common issues of alignment between the display and the moving eye’s pupil. It will enable the stimulation of any retinal location in a controlled way (e.g., in shape, motion, and color), facilitating the establishing of cognitive correspondence.

## Figures and Tables

**Figure 1 sensors-22-06731-f001:**
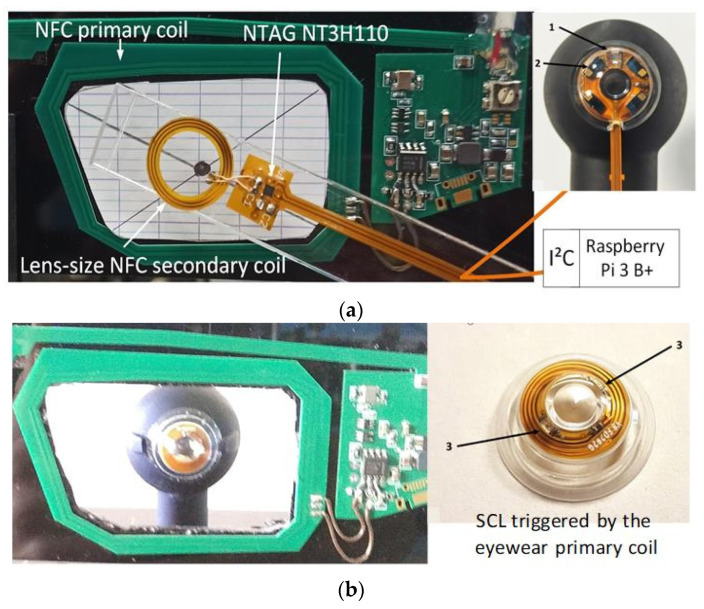
SCL with the eyewear: (**a**) SCL encapsulating four photodiodes (1), an ASIC and a communication module (2); (**b**) SCL encapsulating two VCSELs (3) used as a laser pointer.

**Figure 4 sensors-22-06731-f004:**
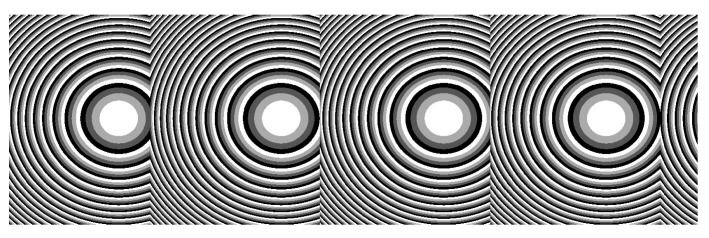
Example of DOE using an off-axis Fresnel lens pattern (i.e., focusing with an axial shift that can be integrated) for four positions.

**Figure 5 sensors-22-06731-f005:**
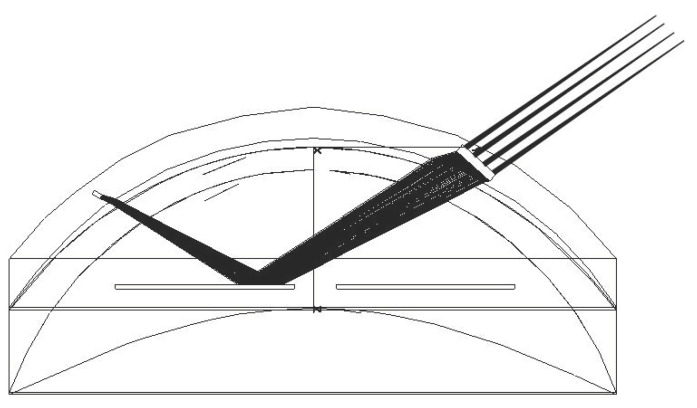
Example of the DOE mode of use when the sensor is outside the lens. Here, four micro-lenses form one to four focal spots, depending on the size of the pupil and thus of the iris surface illuminated.

**Figure 6 sensors-22-06731-f006:**
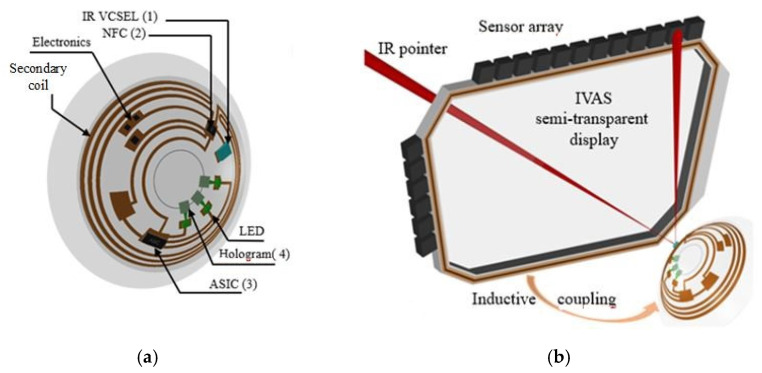
(**a**) SCL configuration when the sensor is not embedded in the lens. (**b**) Several 1D arrays (vertical) could be assembled horizontally (forming a 2D matrix) to include horizontal eye rotation.

**Figure 7 sensors-22-06731-f007:**
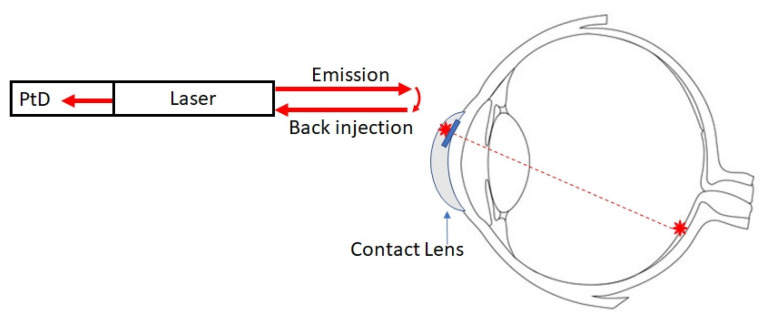
Refractive error measurement by self-mixing interferometry.

**Figure 8 sensors-22-06731-f008:**
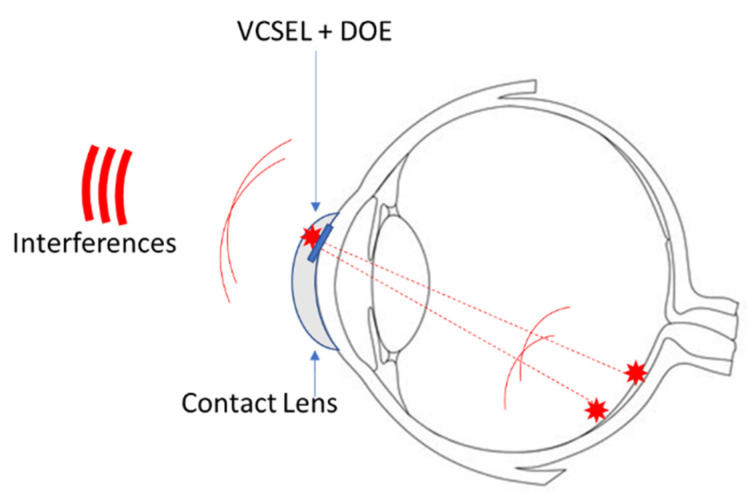
Refractive error measurement by interferometric method.

**Figure 9 sensors-22-06731-f009:**
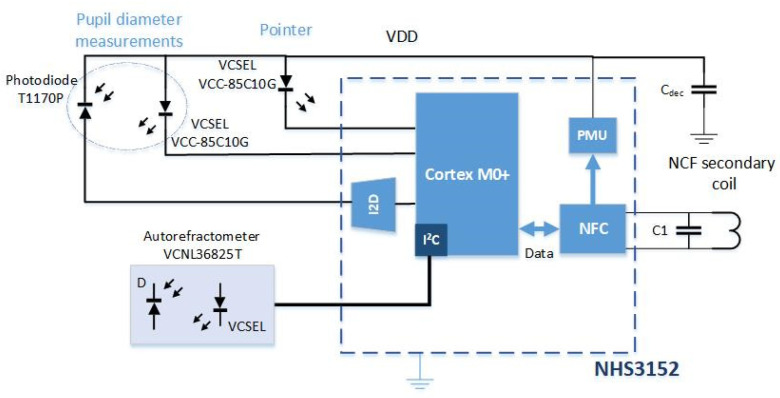
SCL simplified block diagram with the three functions for on-lens computation, built around the NHS3152 smart sensor IC from NXP.

**Figure 10 sensors-22-06731-f010:**
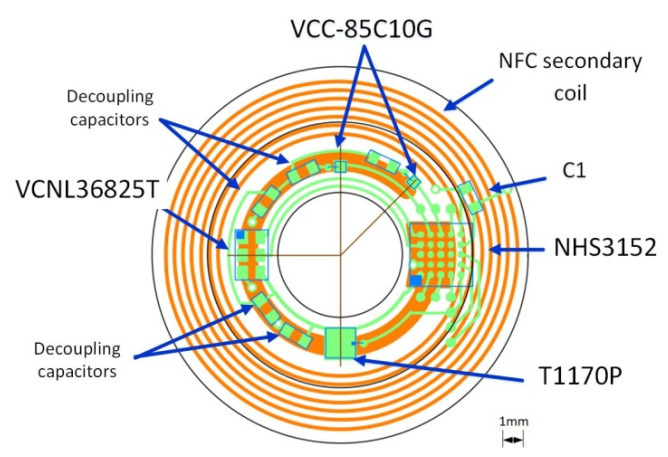
Implementation of the three functions of the block diagram of Figure 9 (VCC-85C10G: VCSEL; T1170P: photodiode; VCNL36825: proximity sensor as refractometer; NHS3152: smart sensor IC).

**Figure 11 sensors-22-06731-f011:**
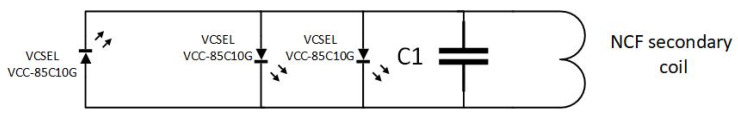
SCL simplified block diagram with the three functions for remote computation.

**Figure 12 sensors-22-06731-f012:**
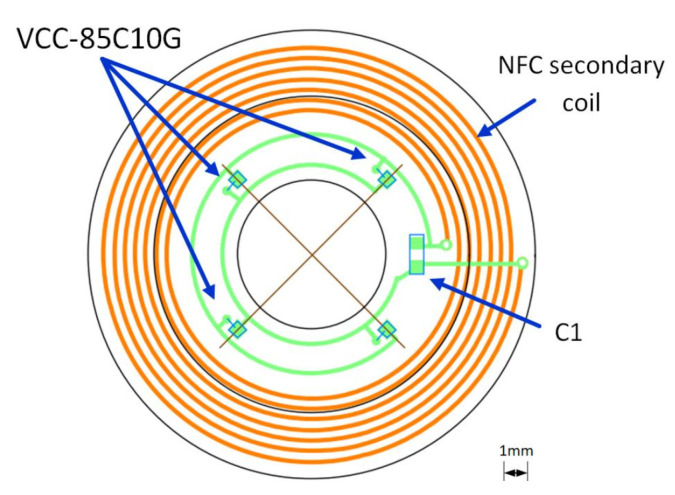
Option with remote computation (VCC-85C10G: VCSEL).

**Figure 13 sensors-22-06731-f013:**
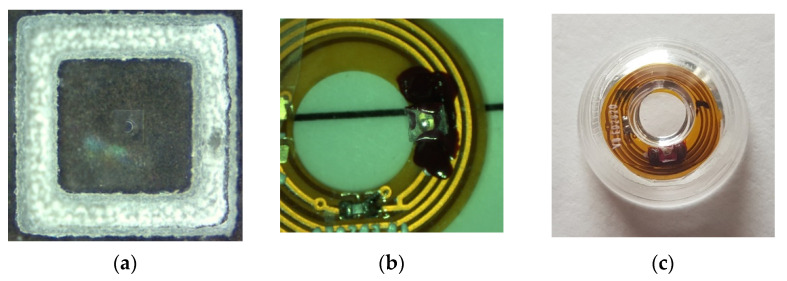
(**a**) DOE of dimensions 225 × 225 µm^2^ on a glass square of 1.2 × 1.2 mm^2^, (**b**) mounted on the VCSEL, (**c**) encapsulated in the SCL [20].

**Table 1 sensors-22-06731-t001:** Multimodal SCL parts’ description.

Function	Part	Manufacturer	W (mm)	L (mm)	H (mm)
Integrated circuit	NHS3152 (WLCSP25)	NXP Semiconductor, Eindhoven, The Netherlands	2.51	2.51	0.5
Photodiode	T1170P	Vishay, Malvern, PA, USA	1.17	1.17	0.28
850 nm VCSEL	VCC-85C10G	Lasermate Group, Walnut, CA, USA	0.25	0.25	0.15
Proximity sensor	VCNL36825T	Vishay Semiconductor, Malvern, PA, USA	2	1.25	0.5

## Data Availability

Not applicable.

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
