# Peer review of "Design of a Multimodal Oculometric Sensor Contact Lens"

_sensors, 2022, doi:10.3390/s22186731_

Round 1

Reviewer 1 Report

I enjoyed reading the paper. The authors have presented an interesting article about a design of a multimodal oculometric sensor contact lens.

Abstract, overview
The abstract is a concise description of the work. The introduction is well structured, and it covers all the concepts investigated in the methodological part. The previous work is well presented and integrated. I consider that this work brings added value in the field and the specific objectives of the manuscript are well related to the previous work developed in this domain. 

Methodology
The research design used is appropriate in order to answer the research questions proposed by the authors. The methods are described properly. The results are clearly presented and are in relation to the concepts investigated.

Discussion and conclusions
The discussions are clear and concise. The conclusions are strongly related to the findings of the research work.

Format and style
All the format and style features were respected and are compliant with the requirements.

References
The format of the reference list fixes well to the specified format.

Plagiarism and any other ethical concerns about this study
I do not have any potential conflict of interest with regards to this paper.

Despite the good work done, there is still some room for improvement, as follows:

  • I think some more literatures should be added. Besides the mentioned HMI there are several other systems (like cost-effect BCI, eye-tracking) which are applied nowadays. It would be good to see the "effect of different web-based media" content on "human brain waves", as well as the additional applications of brainwave-based control like in examine the effect of different web-based media on human brain waves. It would improve the quality of the publication to mention the relationship between a cognitive psychological attention test and the attention levels determined by a BCI systems such as in an examination and comparison of the EEG based attention test with CPT and TOVA. In addition to BCI systems, mentioning other important human-computer interaction eye movement tracking would also improve quality, as such systems can be used in the analysis of programming technologies such as LINQ and algorithms, thus enabling, for example, cognition load or source code, algorithm description tools readability testing like in measuring cognition load using eye-tracking parameters based on algorithm description tools, in clean and dirty code comprehension by eye-tracking based evaluation using GP3 eye tracker and in analyse the readability of LINQ Code using an eye-tracking-based evaluation.

Author Response

To address your comments we have first modified the last part of the first paragraph of the introduction as such:

"... for a wide variety of tasks from achieving basic tasks such as selection, manipulation, navigation [8-9], to assessing mental workload [10] passing by probing memory [11] or analysing programming technologies [12]."

References [8-10] and [12] have been added.

The second paragraph of the introcduction has been extended too as such:

"...More generally, such kind of devices is a key element for investigating the impact of multitasking which is characteristic of the use of mixed reality systems on our cognitive performance, complementary to other cost effective Brain Computer Interfaces (BCI) such as described in [14]."

Reference [14] has been added too.

Finally, as some references were not used, we have redone the reference section

Reviewer 2 Report

The work mainly analyzes the realization and integration of three modal functions on SCL (smart contact lens) while preserving user mobility and discusses issues such as device embedded circuitry, package volume, and energy. The functions of the SCL include measuring gaze direction (GD), measuring pupil diameter (PD), and accommodation changes (AP)which can be applied in IVAS (integrated visual enhancement system). There are excellent application prospects.

The first part analyzes two options to realize the SCL function, one is to detect and calculate the SCL and then transmit it to an external device, and the other is to extract and analyze it through an external sensor.

The second part starts with the study of detecting gaze direction. The author proposes two ways: to emit a laser on the glasses, and the PID (photodiodeidentification of the SCL is used. The rotation of the human eyeball causes the amount of incoming light to change and affects the current of the PID, and the current signal is converted into a digital signal output in the ASIC ( integrated circuit). 

Then came the study of detecting pupil diameter. Depending on the internal structure of the eye, light is reflected off the iris and onto the pupil device, and it continues into the lens to the retina. So to measure the size of the pupil, the diameter of the pupil can be obtained by emitting light to the iris to detect where the demarcation point is. The first necessary equipment is the infrared laser and optics on the SCL. The author proposes two ways. The first is to emit light to the iris and pupil through the laser on the SCL and reflect it to the sensor on the other side of the SCL for identification, and the chip calculates the pupil diameter and outputs it through NFC. The second is linked to sensors on the glasses. SCL The lasers on the iris and pupil emit light to the glasses.

Lastly is the study of accommodation changes (automatic refractometers ). This part is similar to the problem of being out of focus in nearsighted or farsighted situations, where light cannot reach the retina. There are two methods of self-mixing interferometry and interferometry. In SCL, the laser is emitted to the retina for light interference, and then the interference pattern is obtained for analysis, and the output power is obtained to obtain the phase difference and equal reflection for analysis. The first is to detect the self-interference image of both emitted and reflected waves by means of a photodiode etc., on the SCL, which is measured at the SCL level. The second is that the laser can obtain two points through the DOE. After the reflection of the two light points, the interference pattern obtained by coherence is measured and analyzed by the sensor outside the glasses.

The third chapter is to study how to fuse the three detections in SCL. First, some required configurations, three lasers, a contact acquisition system, etc., are mentioned for the fusion of the three functions, and the vergence value of the eye related to the measurement of accommodation ability is also noted to illustrate the role and necessity of accommodation ability detection. Finally, the two limiting factors of volume and energy source when encapsulating in SCL are mentioned. Of course

The specific circuit, hardware, and arrangement of the realization of the multimodal function SCL are explained in detail. Press whether in SCL is two types of hierarchical calculations. The more complicated one is mainly the first one. When calculating SCL, it is necessary to consider the packaging space and energy design of each component. On the SCL, two lasers and a PTD are connected through an IC, one laser emitting inwards and PTD recognition to measure pupil diameter, and one laser emitting outwards to measure gaze direction; there are also lasers and diodes packaged in proximity sensors to test the adjustment ability. The second is just to put the position of the laser on the line, and the sensor is concentrated on the outside. At this point, two ways to implement the function of three-modal detection on SCL have been demonstrated.

In the discussion, the two methods are compared, and the scalability and versatility of the second method are affirmed, but its inconvenience is also explained. It also mentioned the future application of DOE in the second method, the expectation for PRAR, and explained the corresponding advantages of the three functional detections.

Some current eyewear applications, such as Sony's smart contact lenses, have sensors that detect blood sugar in tears and give feedback through color. Or smart contact lenses that can display the content of the user's phone, capture images, and control the movement of the mouse cursor on the screen through the user's eyeballs, but this is achieved by eye tracking with an external camera, which is accurate compared to the method in the text. Higher degree, the realization of a single function is similar to the technology of Sony's smart contact lenses.

As mentioned in the text, SCL has its energy source, can operate and wirelessly communicate, perform actions such as opening or closing, and transmit data to external monitoring equipment. The glasses in this article can be extended to many products, such as VR glasses, mixed reality headgear, smart glasses, etc. After adding SCL, the movement of the eyeball can be reflected in the operation interface in the form of a cursor; or by observing the characteristics of human eye movement, It can also provide visual guidance to reduce cognitive load and establish links between overall planning, control functions, and sensory coordination. In particular, under the trend of virtual, augmented, or mixed reality and other technologies, it is very important to establish a future interaction mode that is mainly based on the eyes and leads to the coordination of the whole body. The content presentation and recognition on display can be provided and expanded externally and using the control terminal attached to the human body itself as the main body must not only be able to recognize the corresponding actions of the eyes but also give feedback to the external device according to the surface changes of the human eyes. middle.

For the two implementation methods in this article, the first type of people can change different peripherals at will to obtain different experiences, but the problem is that it is limited by the space, weight, and energy of SCL, and its computing power is different from more The processing of eye movement features appears to be insufficient. The second method is relatively more convenient in SCL due to more peripherals and more complex design, just adding more sensors. The SCL just acts as a transmitter, continuously transmitting information to the external sensor, but it has no effect once it is separated from the peripheral. If you have to bring a peripheral device when you use it, you can get a new interactive world by limiting the visual range of the eyes or increasing the burden on the body, and the distance between the peripheral device and the eyes is very limited. If the devices in the surrounding environment can be identified and communicated, the first method is to use the operating unit according to one's own eyes in the interface of the external environment and transmit signals through NFC to achieve the purpose of controlling the environment.

By the way, there are some errors in the text. For example,

Page 2 , 2.1 Line 90: CSL?

Page 10 , 3.2, Line 333 : Figure 14, Figure 13(a)

To conclude, I like this work, but it requires additional improvements before being accepted for publication.

Author Response

You are right, the second option requires better knowledge on the eye movements to provide an accurate and reliable information on the oculometric data.

Therefore, we have added a new paragraph to section 

3.2 Importance of the addition of DOE

"Compared to the on-lens computing case, this option assumes a better knowledge on the eye movements to provide an accurate and reliable information on the oculometric data. Even if the eye motions are in practice limited within a solid angle of less than 15° a data processing of the various measurements obtained from the different sensor sources using for instance deep learning techniques (including, for instance, knowledge on eye motion common trajectories and saccades) would be useful to make our system more robust."

We have corrected several typos/errors besides those you pointed out

Round 2

Reviewer 1 Report

I accept in present form.